# Atomic force microscopy methodology and AFMech Suite software for nanomechanics on heterogeneous soft materials

Massimiliano Galluzzi [1,2], Guanlin Tang[1], Chandra S. Biswas[1], Jinlai Zhao[1,3], Shiguo Chen[1] & Florian J. Stadler [1]

Atomic force microscopy has proven to be a valuable technique to characterize the mechanical and morphological properties of heterogeneous soft materials such as biological specimens in liquid environment. Here we propose a 3-step method in order to investigate biological specimens where heterogeneity hinder a quantitative characterization: (1) precise AFM calibration, (2) nano-indentation in force volume mode, (3) array of finite element simulations built from AFM indentation events. We combine simulations to determine internal geometries, multi-layer material properties, and interfacial friction. In order to easily perform this analysis from raw AFM data to simulation comparison, we propose a standalone software, AFMech Suite comprising five interacting interfaces for simultaneous calibration, morphology, adhesion, mechanical, and simulation analysis. We test the methodology on soft hydrogels with hard spherical inclusions, as a soft-matter model system. Finally, we apply the method on *E. coli* bacteria supported on soft/hard hydrogels to prove usefulness in biological field.

[1] College of Materials Science and Engineering, Shenzhen Key Laboratory of Polymer Science and Technology, Guangdong Research Center for Interfacial Engineering of Functional Materials, Nanshan District Key Lab for Biopolymers and Safety Evaluation, Shenzhen University, 518055 Shenzhen, China. [2] Shenzhen Key Laboratory of Nanobiomechanics, Shenzhen Institutes of Advanced Technology, Chinese Academy of Sciences, Shenzhen University Town, 1068 Xueyuan Avenue, 518055 Shenzhen, Guangdong, China. [3] Faculty of Information Technology, Macau University of Science and Technology, Avenida Wai Long, 999078 Taipa, Macau, China. Correspondence and requests for materials should be addressed to F.J.S. (email: fjstadler@szu.edu.cn)

Atomic force microscope (AFM) evolved from being a pure mechanical microscope, being able to operate in diverse environments (vacuum, air, or liquid), on a multitude of different specimens, to a highly sophisticated toolbox with the possibility (among others) to correlate morphology on nanoscale with several physico-chemical properties, like adhesion, friction, and elasticity. Special efforts were devoted to the investigation of mechanical properties of soft-matter such as soft polymers and biological materials (biomembranes, bacteria, living cells, and tissues)[1,2]. Biological functions of living cells, as well as their pathophysiological state, can be correlated to local mechanical properties, for example, discriminating between healthy and cancer cells[3–6]. Moreover, AFM allows for deeply investigating how mechanical properties of surrounding environment are influencing the behavior of cells[7–9]. This leads to several biomedical applications, such as regenerative medicine and tissue engineering[10,11]. Only recently[12,13], several research groups devoted themselves finding a common methodology and protocol in order to validate, quantify, and ensure repeatability of results obtained in different laboratories, having led to the standardized nanomechanical AFM procedure (SNAP), introducing a methodology and relative corrections developed within a large network of laboratories[13]. While this is an important leap, the state-of-the-art methodology cannot overcome the intrinsic complexity of cells and biological world in general: strong heterogeneity from nanoscale to microscale (external and internal), morphological features, and natural time-dependent dynamics are raising challenges in experimental execution and particularly in data analysis. Moreover, standard theoretical models used for nanomechanics data analysis fail to describe the heterogeneity of biological systems due to non-neoHookean behavior and/or microscopic heterogeneity, often requiring advanced computational modeling[14–16].

In this work, we propose a methodology in order to investigate heterogeneous specimens following three steps: (1) calibration of AFM setup through the SNAP method and comparison with macroscopic rheology on homogeneous hydrogels. (2) AFM space-resolved nanomechanical experiments using spherical colloidal probes (the procedure also works with standard sharp probes but leading to inferior accuracy and possible damage to living cells due to the sharp apex of the indenter[17,18]). (3) Direct comparison of AFM single force spectra with axisymmetric finite element simulations (FEM) based on AFM mapping, leading to several advantages: accurate representation of complex systems (dividing geometries and heterogeneous materials in small connected parts, so-called mesh) with a visual representation of the overall representation, still capturing local effects. The overall protocol is integrated into a custom freely available software developed with the idea to easily manage raw data analysis to obtain reproducible and quantitative results. AFMech Suite is composed of five interacting Matlab-based graphical user interfaces in order to perform base or advanced AFM analysis from probe calibration to final results in a real-time and overall user-friendly environment, allowing for direct comparison with external data including FEM simulations. Several contact mechanics models are available for data analysis in order to consider different probe geometries (sphere, cone-pyramid, cylinder, hyperboloid) and sample conditions (adhesion, non-linearity). The software is built to expanding the custom analysis routine developed for nanomechanical analysis of living cells[19].

## Results

### Rheology and AFM from macro to micro.
Calibration of AFM setup is crucial in obtaining quantitative nanomechanical

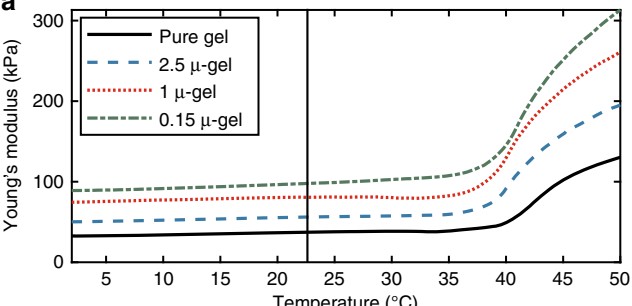

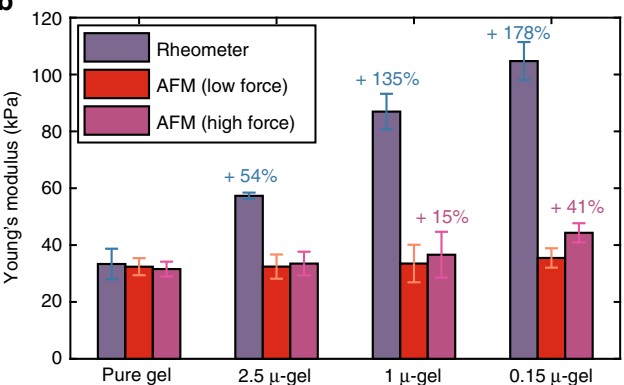

**Fig. 1** Comparison between microscopic and macroscopic Young's modulus. **a** Rheological shear modulus measurements converted to Young's modulus using $n = 0.5 \rightarrow E = 3G$ on composite hydrogel series with temperature sweep. Data were converted from shear modulus to Young's modulus for direct comparison with AFM results. **b** Rheometer results in comparison with spherical colloidal probe AFM analyzed at low (0.2–20 nN) and high (40–60 nN) force in sufficient distance from visible particles. Error bars represent the quadrature sum of single measurements uncertainty and standard deviation of 10 replicates obtained at different macroscopic positions

properties with high accuracy. For this purpose, a series of homogeneous hydrogels was produced and used recursively applying SNAP method[13] to obtain correct AFM calibration parameters. In the case of heterogeneous gels, the comparison between microscopic AFM and macroscopic rheology is not less important; however, considerable deviations are expected. While rheology measures averaged mechanical properties of overall sample with very high accuracy, AFM can spatially resolve them depending on the ratio between size of the probe and heterogeneity scale length. Temperature-dependent Young's moduli from rheological measurements (angular frequency $\omega = 1\,s^{-1}$, heating rate $q = 1\,K\,min^{-1}$, converted from shear moduli using the Trouton ratio[20]) for pure gel, 2.5 μ-gel, 1 μ-gel, and 0.15 μ-gel, where the number stands for the radius of the polystyrene (PS) sphere diameter in μm (concentration in swollen state, $\phi_F = 0.4$ wt.% in swollen state as used for the AFM testing, see Supplementary Table 1), are presented in Fig. 1a and numerical data in Table 1. Young's modulus increases with decreasing particle size, which has been observed several times in rheology before (observing this effect proved to be more complicated than expected at a first glance as especially for nanoparticles, aggregation, and consequent entrapping of polymer between the particles can have a severe effect on the rheological behavior).[21,22]

Standard classic models describe the modulus of composite material only dependent by filler concentration leading to discrepancies with our observations in Fig. 1. Kerner's equation[23]

**Table 1 Numerical values of moduli for composite gels along with theoretical parameters**

| Sample | <d> (μm) | Apparent $\Phi_F$ Kerner | Rheology (kPa) | AFM low (kPa) | AFM high (kPa) |
|---|---|---|---|---|---|
| Pure gel | \\ | \\ | 33.4 ± 1.4 | 32.4 ± 3.0 | 31.6 ± 2.6 |
| 2.5 μ-gel | 14.06 | 22% | 57.3 ± 1.1 | 32.4 ± 4.2 | 33.5 ± 4.1 |
| 1 μ-gel | 5.62 | 39% | 86.9 ± 3.2 | 33.5 ± 6.1 | 36.7 ± 7.1 |
| 0.15 μ-gel | 0.844 | 46% | 104.7 ± 6.7 | 35.4 ± 3.4 | 44.3 ± 3.4 |

Errors in rheological tests represent the standard deviations obtained on 4 different replicates, while errors for AFM represent the quadrature sum of single measurements uncertainty and standard deviation on 10 replicates obtained at different macroscopic positions

predicts an increase of modulus around 1 % (from 33.4 to 33.8 kPa) for $\phi_F = 0.4$ wt.% and filler modulus $E_F = 3.3$ GPa[24].

$$E_C = E_M \frac{(1 - \phi_F)E_M + (\alpha + \phi_F)E_F}{(1 + \alpha\phi_F)E_M + \alpha(1 - \phi_F)E_F} \qquad \alpha = \frac{2(4 - 5\nu_M)}{(7 - 5\nu_M)},$$

(1)

where $E_C$, $E_M$, and $E_F$ represent the Young's modulus of composite, hydrogel matrix, and filler, respectively, $\alpha$ is a parameter depending on Poisson's ratio $\nu_M$ of matrix phase ($\nu_M = 0.5$ for incompressible solids and liquids). PNIPAM-based hydrogels behave as almost ideal rubbers and, due to the incompressibility of both polymers and Newtonian liquids like water, can be treated as incompressible materials. Previous experimental works from our group comparing AFM indentation with rheology confirmed this Poisson's ratio assumption[17,25–27], where Cox–Merz rule[28] (linking dynamic-mechanical and start-up shear experiments) was used to convert shear into elongational modulus by $3|G^*| = E$. Finally, previous direct rheological data demonstrate incompressibility behavior[29]. Inverting Eq. (1), the macroscopic moduli obtained by rheology would expect an apparent filler concentration between 20 and 50 wt.% (see Table 1), which can be explained by the fact that the Kerner model does not take into account the interface between hard fillers and surrounding hydrogel matrix, thus neglecting surface effects. Classic models, depending only on volume fraction of filler, are good approximations of mechanical properties for composite with millimeter or micrometer sized fillers[30], but fail to describe nanocomposites when surface interactions become dominant. Lewis and Nielsen observed that modulus increases as the particle size decreases, stating how the increased surface area provides a more efficient interfacial bond[31]. In our case, segmental immobilization caused by the interaction of PNIPAM polymer hydrogels chains with the filler surface supports the idea of enhancement in structural reinforcement and Young's modulus[32,33]. Similar effects were analyzed by Alimardani et al.[34] proposing a 3-phases (matrix, filler, and interphase) model of composite where volume and surface filler effects are separated. In that work, the increase of stiffness resulted from the chain confinement of rubbers by filler aggregates; in particular, the interphase between filler and matrix is acting as a transition zone with mechanical properties between those extremes[34]. As shown later, these considerations are helpful in designing the interfacial boundary behavior of the FEM model.

Moreover, AFM was used to statistically analyze (10 measurements each sample) the matrix phase around inclusions. This analysis was possible by creating a morphological mask with AFMech Suite excluding and avoiding particles and morphological inhomogeneity. As shown in Fig. 1b, AFM with colloidal probe on surrounding hydrogel matrix deviates from rheology, showing values similar to that of the pure gel. This behavior is expected as AFM indentation has intrinsically more spatial resolution than rheometer allowing to distinguish fillers from surrounding matrix. Especially for shallow indentation (low force) no variation is detected, while for 0.15 μ-gel at deep

indentation and to a smaller degree for the 1 μ-gel (high force) the Young's modulus increased, although still underestimating rheological values. This is correlated with the intrinsic resolution of AFM and heterogeneity length scale of specimen under indentation. If the probe size and indentation length are larger than the length scale of different phases in the composite, AFM resolution will not be enough to resolve moduli for single phases resulting in averaged estimations of matrix and particle-influenced matrix. Scale lengths relative to AFM nano-indentation are connected to the lateral resolution dependent on contact area between probe and sample. In our experiments, typical radii of contact area depend both on probe radius ($R$) and indentation length ($\delta$), leading to $a = (\delta R)^{0.5} = 1.6$ μm, where $R = 2.5$ μm and $\delta = 1$ μm. Scale lengths relative to the heterogeneity of specimens depend on geometrical distribution of different phases. For the specimens in this study, the statistical distribution of randomly dispersed spheres was calculated[35], formulating the average nearest-neighbor separation as:

$$\langle d \rangle = 0.554 \sqrt[3]{\frac{4\pi}{3\phi_F}} R_P,$$

(2)

where $R_P$ represents the radius of spherical inclusions. The numerical values of average nearest-neighbor separation are listed in Table 1. The increase of Young's modulus measured by AFM for high force is explained by radius of contact area in the same range of nearest-neighbor separation; thus, the probe starts to sense averaged mechanical properties. This effect of local averaging is particularly useful to investigate the mechanical properties of cytoskeleton in living cells. Micrometric spherical probes are selected specifically to average local features on biomembranes such as glycocanes, brushes, and membrane proteins, but maintaining enough spatial resolution to focus on mechanical response of cytoskeletal layers (i.e., actin and microtubules)[17,19].

Several applications already exploited nanocomposite hydro-gels, for example, in electronics, biosensing, catalysis, and drug delivery and regenerative medicine[36]. Not only the hydrogel matrix is acquiring or enhancing different properties, but also composite structure provide the means to control nanoparticles aggregation avoiding environmental leaking (for nanotoxicological purpose). Several reviews on this topic have been published recently[36–41].

**AFM mechanical imaging and simulations.** AFM results about indentation on single spherical particles exposed from hydrogel matrix level for 0.15 μ-gel (Fig. 2), 1 μ-gel (Fig. 3), and 2.5 μ-gel (Fig. 4). Figure 2 evidences a partially embedded PS sphere residing at the top of hydrogel matrix level. Lateral size of inclusion in Fig. 2a is overestimated because of convolution effect with colloidal probe ($R = 2.5$ μm). The heterogeneity in Fig. 2b–d depends on AFM tip, after vertical contact, transferring force to the hard inclusion, which is then acting as an apparent new indenter. Deviations from Young's modulus of surrounding

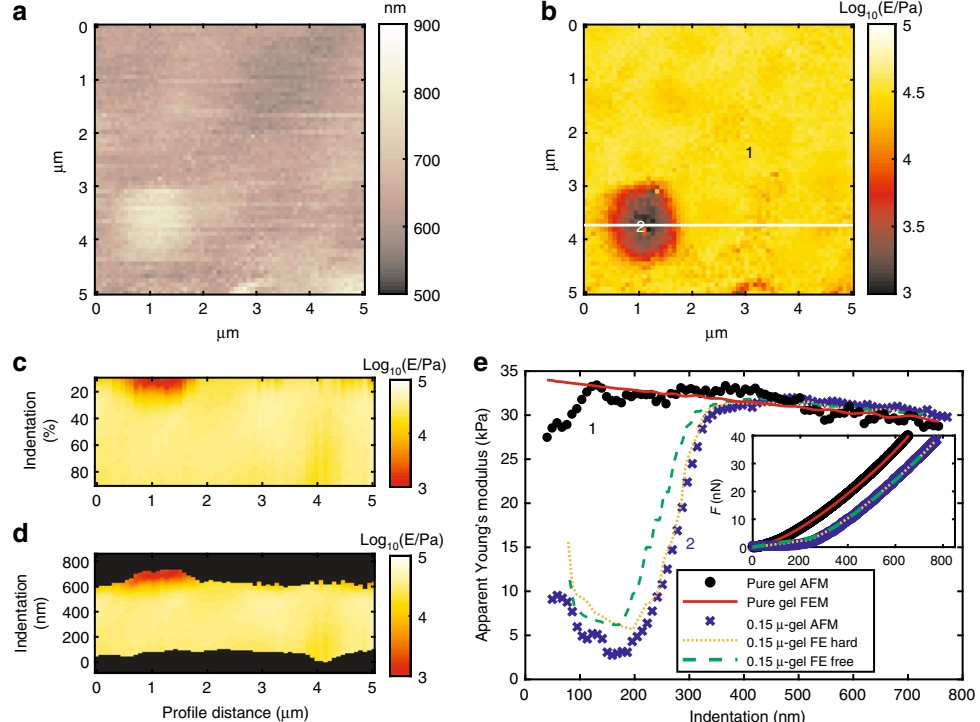

**Fig. 2** Graphical results of mechanical analysis by AFMech Suite for 0.15 μ-gel. **a** Morphology map at zero force, vertical color bar represents height in nm. **b** Young's modulus map in logarithmic scale at medium indentation (0–20% of maximum value). **c** Qualitative Young's modulus tomography on relative indentation from white line in (**b**). **d** Qualitative Young's modulus tomography on absolute indentation from line in (**b**). Vertical color bar for (**b**–**d**) represents logarithm base 10 of Young's modulus in Pa. **e** Apparent Young's modulus vs. indentation (inset: force vs. indentation) for points 1 and 2 of (**b**) comparing AFM results and FEM simulations for boundary conditions: no slip (FE hard) and free to slip (FE free)

matrix are expected and caused by a combination of mismatches of contact area and indenter radius when fitting using standard contact mechanics model (Hertz model in this case). Analysis was performed by dividing the indentation curve in small intervals and fitting them using Hertz model in order to appreciate the variability of apparent Young's modulus. Apparent Young's moduli vs. indentation evaluated at the top of exposed sphere, shown in Fig. 2c–e, can be summarized in three parts: (1) Initial overestimation of Young's modulus due to force transferred to inclusion having non-zero contact area on gel matrix, (2) starting contact area becomes negligible in comparison with new contact area experienced by filler indenting gel: apparent modulus responds depending on radius of probe and filler (see Eq. (3)), (3) at large indentation apparent modulus reach matrix value for filler smaller than probe, or remain overestimated for filler bigger than probe. In the second trend, when filler indenting the gel effect dominates on starting contact area, the apparent modulus is obtained considering the spherical geometry and radius of the filler, as depicted in Eq. (3):

$$E_{\text{app}} = \frac{\sqrt{R_{\text{fil}}}}{\sqrt{R_{\text{tip}}}} E_{\text{gel}}, \tag{3}$$

where $E_{\text{app}}$, $E_{\text{gel}}$, $R_{\text{tip}}$, and $R_{\text{fil}}$ are, respectively, apparent Young's modulus, Young's modulus of surrounding hydrogel, radius of colloidal probe, and radius of spherical filler. The equation trend was confirmed by FEM previously[25]. The behavior for large indentation depends also on the ratio between sizes of probe and filler: for smaller fillers, modulus is reaching matrix value because the probe is dominantly in contact with gel matrix. Also in this case, the contact area between probe and gel is the key in interpretation of effects as long as most of the probe surface area is in

contact with hydrogel and filler effect is negligible. For big filler particles, the probe is unable to contact the gel causing a constant overestimation of apparent modulus and following Eq. (3).

The three trends are clearly visible for 0.15 μ-gel in Fig. 2e, especially the second trend showing apparent Young's modulus predicted by Eq. (3): $E_{\text{app}} = 0.24$, $E_{\text{gel}} \approx 8$ kPa. This is a typical situation of heterogeneous system where standard contact mechanics models can hardly be used, while FEMs will acquire a primary role for data interpretation. An array of simulations was built, starting from parameters measured by morphology and mechanical analysis from AFM and rheology. Moreover, the system studied here was maintained as simple as possible using spherical filler with well-defined size and shape with a much higher modulus than the hydrogel matrix. Therefore, the amount of free parameters is minimized to only comprising vertical position of the particle to the surface of the sample and particle–matrix interface behavior. For simulating the experimental particle–matrix behavior by AFM two limiting cases have to be considered: free-slip or zero-slip. When comparing both FEM simulations (green and orange line in Fig. 2e) with the AFM data (dots), a smaller discrepancy is found for the zero-slip case, suggesting that the hydrogel matrix does not slip (very much) around the particle. Thus, the bond between matrix and filler is stronger than the indentation force. While this cannot be directly validated by macroscopic rheology, the fact that the modulus increases significantly due to chain immobilization on the particle surface, is a strong support of the non-slip assumption, as slip would mean that the chains are not or only slightly immobilized.

A similar behavior was observed for 1 μ-gel in Fig. 3: the first trend relative to mismatch of starting contact area is visible in Fig. 3c–e, along with the second trend related to filler indenting the matrix predicted by Eq. (3): $E_{\text{app}} = 0.63$, $E_{\text{gel}} \approx 18$ kPa. Due to

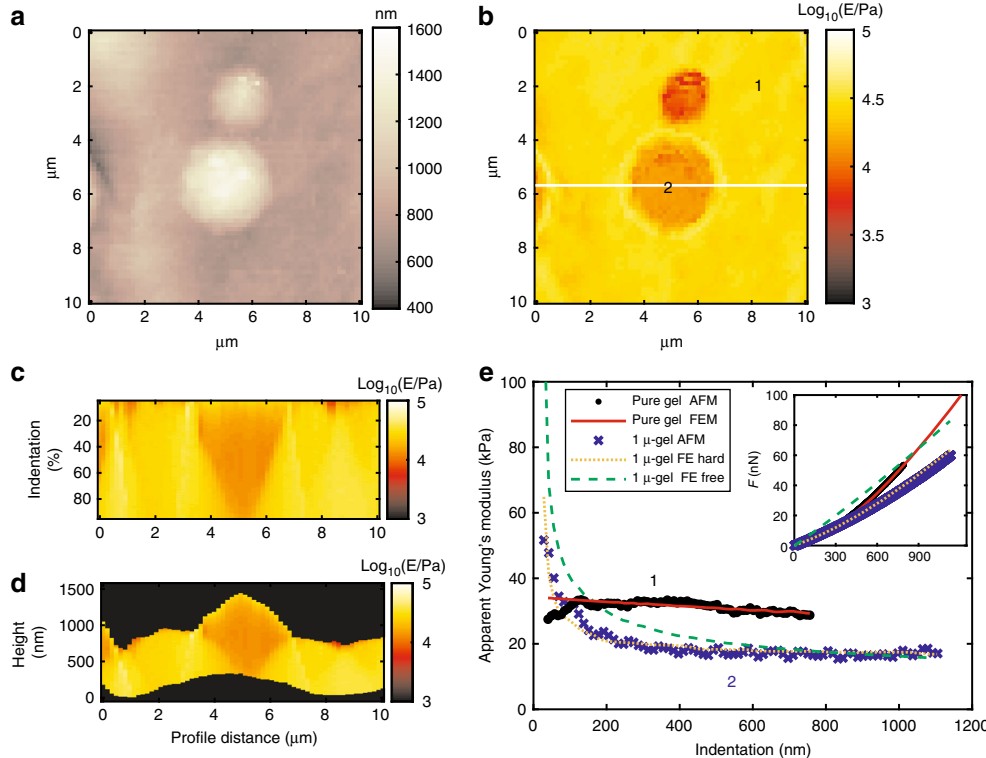

**Fig. 3** Graphical results of mechanical analysis by AFMech Suite for 1 μ-gel. **a** Morphology map at zero force, vertical color bar represents height in nm. **b** Young's modulus map in logarithmic scale at medium indentation (20–40% of maximum value). **c** Qualitative Young's modulus tomography on relative indentation from white line in (**b**). **d** Qualitative Young's modulus tomography on absolute indentation from line in (**b**). Vertical color bar for (**b**–**d**) represents logarithm base 10 of Young's modulus in Pa. **e** Apparent Young's modulus vs. indentation (inset: force vs. indentation) for points 1 and 2 of (**b**) comparing AFM results and FEM simulations for boundary conditions: no slip (FE hard) and free to slip (FE free)

the larger particle size, however, maximum AFM indentation (≈1 μm) is not large enough to visualize the third trend when apparent modulus approaches matrix value; still FEM simulation is confirming the trend at 2.5 μm (data shown in Supplementary Fig. 8). If the probe is approaching the particle near the edges, a third trend is becoming visible due to the probe laterally shifting the inclusion, thus enhancing the contact area with the gel. Also for 1 μ-gel sample, the hydrogel matrix is strongly bound to the filler surface (Fig. 3e, zero-slip simulation (orange line) matches experimental AFM data (dots) well).

The mechanical properties of sample 2.5 μ-gel are mainly dominated by starting contact area effect (first trend) causing an increasing of apparent Young's modulus (Fig. 4b–d) by up to 1 order of magnitude. Apparent modulus is progressively decreasing with indentation (Fig. 4e), indicating that filler proceeds to indent the matrix (second trend). The filler has the same size as the probe; therefore, the apparent modulus could reach $E_{app} = E_{gel} \approx 30$ kPa only asymptotically, especially visible from FEM simulated curve. Also in the case of 2.5 μ-gel, the preferred boundary condition is zero-slip, again underlining the strong bond between PS sphere and hydrogel.

**Application to biological specimens.** The complete 3-step procedure (calibration, AFM indentations, and FEM simulations) was applied to a biological system after successful validation on above described PS sphere filled hydrogels. *E. coli* bacteria were attached to two flat hydrogels with well-defined Young's modulus (soft $E \approx 6$ kPa, hard $E \approx 30$ kPa) and otherwise identical physical properties and chemical composition. AFMech Suite was used

to analyze morphology, dimensions, and modulus of the single gel parts setting the path to retrieving quantitative information with high accuracy for a heterogeneous biological system. Morphologies by AFM nanomechanical measurements (Fig. 5a soft and Fig. 5d hard) show bacteria having their unusual round shape and larger size due to convolution with a spherical colloidal probe ($R = 2.5$ μm) bigger than bacteria size (1–2 μm). Figure 5b, e compares the mechanical maps for single bacteria supported on soft and hard substrates at medium indentation (20–60%), respectively. It is remarkable that on average the bacteria exhibit very different moduli—on the soft hydrogel $E_{bacterium1} = 3.5$ kPa (Fig. 5b) and on the hard hydrogel $E_{bacterium2} = 10$ kPa (Fig. 5d). This apparent difference in the mechanical properties of identical bacteria is the consequence of the combined indentation of the bacterium and its underlying soft substrate, which is an unavoidable consequence of the necessity of providing adequate substrates for many biological samples, especially for eukaryotic cells, for whom substrate modulus might influence specialization. Consequently, the real Young's modulus of the bacterium is always convoluted with the contribution of substrate stiffness leading to artifacts, requiring FEM simulations for proper deconvolution. Center of bacteria in all AFM measurements appears harder, an effect probably due to an accumulation and compaction of genetic material, such as supercoiled DNA located in the cytoplasm and organized in nucleoids[42]. Similar results were obtained by Longo et al.[43], using sharp probes and confirming the association of stiffness by the accumulation of structures lying in the cytoplasm and not by morphological variations of the cells. In our case, colloidal probes are apparently enhancing the perceived

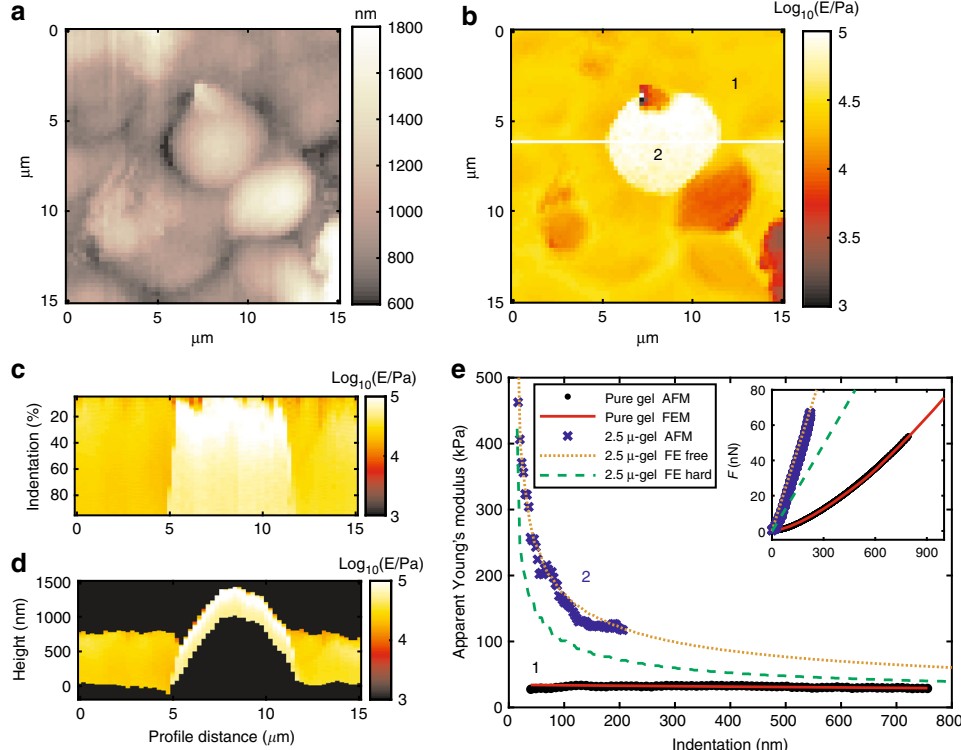

**Fig. 4** Graphical results of mechanical analysis by AFMech Suite for 2.5 μ-gel. **a** Morphology map at zero force, vertical color bar represents height in nm. **b** Young's modulus map in logarithmic scale at medium indentation (40–60% of maximum value). **c** Qualitative Young's modulus tomography on relative indentation from white line in (**b**). **d** Qualitative Young's modulus tomography on absolute indentation from line in (**b**). Vertical color bar for (**b**–**d**) represents logarithm base 10 of Young's modulus in Pa. **e** Apparent Young's modulus vs. indentation (inset: force vs. indentation) for points 1 and 2 of(**b**) comparing AFM results and FEM simulations for boundary conditions: no slip (FE hard) and free to slip (FE free)

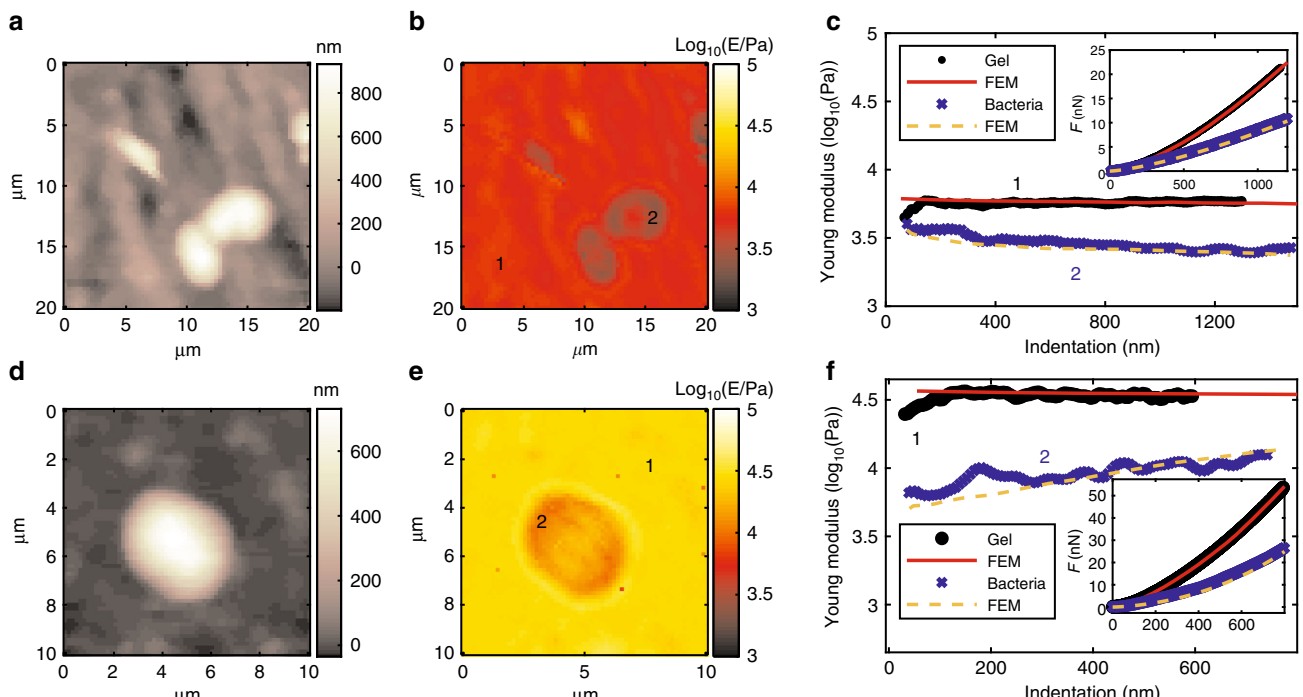

**Fig. 5** Results of mechanical analysis for *E. coli* supported on hydrogels. Bacteria supported on hard hydrogel (6 kPa). **a** Morphology map at zero force, vertical color bar represents height in nm. **b** Young's modulus map at medium indentation (20–40%), vertical color bar represents logarithm base 10 of Young's modulus in Pa. **c** Young's modulus vs. indentation (inset: force vs. indentation) for points 1 and 2 of (**b**) comparing AFM results and FEM simulations. Bacteria supported on hard hydrogel (6 kPa). **d** Morphology map at zero force. **e** Young's modulus map at medium indentation (20–40%). **f** Young's modulus vs. indentation (inset: force vs. indentation) for points 1 and 2 of (**b**) comparing AFM results and FEM simulations

size of such structures. For comparison with simulations, we decided to avoid the central part, as a more complex FEM model (4-component materials) would be required for data interpretation.

As shown in Fig. 5, FEM simulations were related to a 3-component model, previously tested on gels with spherical inclusions, comprising: indenter, bacterium, and underlying gel assuming free-slip conditions on the indenter–bacterium interface. AFM force curves (FC) were chosen on bacterium major axis between harder central part and external region where tip is only displacing partially the bacterium. An array of FEM simulations was produced exploring the elastic modulus of bacteria between 1 and 50 kPa and the most comparable simulation is automatically highlighted in AFMech Suite. Hence, it is concluded that the true stiffness of *E. coli* is 12 kPa independent of the substrate stiffness. When the single bacterium is harder than the hydrogel substrate and smaller than the probe, a situation similar to 1 μm PS inclusion residing on top of hydrogel exists: the bacterium is acting as indenter, therefore leading to counterintuitive mechanical properties. Similar to PS inclusions strongly bonded to the matrix, FEM simulation at zero-slip minimize the discrepancy between is describing the interface between bacteria and gel, indicating a sufficiently strong bond to withstand to indentation forces used. This is in agreement with idea of bacteria starting to generate a strong biofilm on surface, especially when incubated for long time (>24 h). During biofilm formation, bacteria produce molecules and protrusions such as pili, fimbriae, or capsules generating a strong and irreversible bond with the supporting surface[44].

As detailed here, FEM simulations coupled with AFM are necessary in order to deconvolute the contribution of biological object and soft substrate. This analysis, accompanied with "AFMech suite", will be important to remove artifacts and identify quantitatively real differences due to the substrates. The reorganization of cytoskeleton of living cells supported on hydrogels with different stiffness and porosity will be the next extent of this work.

## Discussion

This work introduces a standardized and validated way to carry out nanomechanical characterization of heterogeneous composites systems from hydrogels with inclusions to living bacteria supported by hydrogel surfaces with different mechanical properties. Furthermore, the output of this work regards the realization and validation of well-defined methodology for measurements execution, interpretation, and analysis.

The specific steps of the proposed methodology can be summarized as following:

1. Fine mechanical calibration of AFM through rheological measurements on homogeneous gels, applying recursively SNAP method[13].
2. Systematic analysis of heterogeneous samples (composite gels and biological systems) in physiological conditions using AFM space-resolved indentation.
3. Reconstruction of AFM experiments using FEM parametrized by AFM results[25]. Production of parametric array of simulations for data interpretation.

A customized software was developed in Matlab language and deployed as executable (available in Supplementary Software). The methodology and software were initially tested on a model system composed of hard spheres with known geometry and stiffness randomly dispersed in hydrogel matrix. The system showed a straightforward agreement between AFM indentation and FEM simulations. The no-slip boundary condition between

inclusion and matrix was evidenced by the methodology, furthermore confirming the influence of strong interaction at interface experienced in macroscopic rheology experiments. Finally, the methodology was applied to a biological system, consisting in *E. coli* bacteria supported on hydrogel surfaces with different stiffness. Our protocol leads to the deconvolution of mechanical response of the individual parts of bacterial body, showing no influence of substrate stiffness in the tested range.

## Methods

**Sample preparation.** Details about synthesis of composite hydrogel samples are accurately described in the Supplementary Note 1 and Supplementary Table 1. During synthesis of poly(N-isopropylamide) (PNIPAM) hydrogels, PS monodisperse spheres (with different sizes: $R = 2.5, 1, 0.15$ μm) were added in fixed quantity at 5% relative to monomer weight. After purification and reswelling in water (leading to significantly higher water contents than during synthesis), the samples result in a composite gel consisting in uniform soft matrix embedded with randomly dispersed spheres. The volume filler concentration was calculated as ratio of single phases volumes, therefore $\phi_F = 0.4\%$ for all specimens. Throughout the text, we named the samples as 2.5, 1, and 0.15 μ-gel, respectively. The scanning electron microscopy (described in Supplementary Method 1) of monodisperse spheres deposited on flat silicon (before including in gels) and freeze-dried composite hydrogels are shown in Supplementary Fig. 1.

Gram-negative bacteria *E. coli* (ATCC25922) were purchased from Guangdong Institute of Microbiology, and were incubated on a nutrient agar plate at 37 °C for 24 h before use. Hydrogels with different mechanical properties (6.5 and 27 kPa) were immersed in bacterial dispersion and incubated for 24 h at 37 °C in order to deposit and fix bacteria on the surface. Samples were finally washed and rehydrated with PBS solution, glued to the standard petri dish prior AFM mechanical measurements.

**Calibration of rheology and SNAP.** The rheological experiments were performed using an Anton Paar MCR 302 rheometer, experimental details are described in Supplementary Method 2. Due to the nature of hydrogels being incompressible solids[45] and the validity of the Cox–Merz rule[28], it was possible to convert the magnitude of the complex shear into the elongational modulus by $3|G^*| = E$, thus allowing for comparing shear modulus and AFM modulus.

Geometrical and mechanical calibration of AFM probe are extremely important to achieve quantitative and reproducible results. Spherical colloidal probes (Novascan) with a nominal probe radius $R_P = 2500$ nm and nominal elastic constant $k = 0.06$ N/m were used. The exact geometry of the spherical probe was measured by reverse AFM imaging on calibration grid TGT1 (NT-MDT, see Supplementary Fig. 2). The initial elastic constant was measured by thermal tuning method, after calibration of optical lever sensitivity on hard petri-dish in water environment. During optical lever sensitivity calibration, non-linearity correction for photodetector system was applied systematically (see Supplementary Fig. 3). Young's moduli produced by rheometer was employed to apply the SNAP[13] in order to correct imprecisions during optical lever system calibration. AFM mechanical measurements at microscale, using spherical probes, are in excellent agreement with macroscopic rheology as it was stated in several recent publications of our group[17,26,27]. As external method to measure the spring constant of cantilever we compare moduli from AFM nanomechanics and macroscopic rheology by linear regression method. For this purpose, we created a calibration sample consisting in a series of five mechanically well-defined homogeneous hydrogels showing modulus between 1 and 30 kPa (typical range of biological systems). Because modulus by AFM depends both on elastic constant and lever sensitivity, SNAP procedure was applied recursively (three times) after repeating thermal tuning with new parameters. Additional details and data about calibration and corrections are shown in Supplementary Note 2 and Supplementary Fig. 4.

**AFM nanomechanics.** The local elasticity of composite PNIPAM gels were probed with a commercial AFM Dimension Icon (Bruker) in the force volume (FV) mechanical imaging mode[46]. At every point of a square matrix, a single FC was acquired indenting the surface and producing, after data analysis, a Young's modulus map in 1:1 correspondence with reconstructed morphology. For all measurements, we maintained the total vertical ramp length of FC at $L = 4$ μm, selecting a maximum force $F \approx 60$ nN and vertical approaching/retracting velocity at $v \approx 20$ μm/s. The lateral resolution was fixed at $64 \times 64$, while 4096 points were acquired for each FC. A series of 10 FV measurements were performed in different macroscopic positions to improve the statistical reliability of the experiments. All samples were imaged while they were immersed in deionized water at room temperature ($T = 23$ °C). Optimized parameters for AFM nanomechanics on homogeneous gels were discussed and reported in a previous work[17].

Young's moduli were evaluated by data analysis performed with AFMech Suite software. Raw AFM data were processed to obtain FCs ($F$ [nN]) vs. indentation ($\delta$ [nm]). The non-contact part of FC is initially flattened and aligned at zero force value. Then, vertical force axis was divided in 1000 intervals producing a histogram for each FC: the non-contact part appears as a sharp peak automatically detectable

and fitted by a Gaussian distribution. The region of the FC above experimental noise (1 single width $\sigma$ of the Gaussian distribution) is considered as contact part, defining the positive indentation ($\delta$ [nm]). Total indentation length was used to obtain the real morphology at zero force. For data analysis, we mainly use the approaching curves, an example of complete FC after contact point evaluation is shown in Supplementary Fig. 5c. Extend and retract parts are mostly overlapping demonstrating negligible viscosity (as confirmed by rheology, where we found the phase angle $\delta$ to be close to 0°, which means that the materials are behaving as almost ideal rubbers, i.e., negligible energy dissipation)[29] and negligible adhesion in comparison with maximum applied force. Description and example of adhesion study performed by AFMech Suite is available in Supplementary Note 3 and Supplementary Fig. 5. Despite its simplicity and approximation, we used the Hertz model to retrieve Young's moduli considering the following assumptions: sample must be linear, homogeneous, purely elastic, adhesive force negligible, indentation must be small compared with probe radius and sample thickness. Direct analysis on homogeneous hydrogel using different contact mechanics models is provided in Supplementary Note 4 and Supplementary Fig. 6, along with detailed study on validity of Hertz approximation and paving the road for heterogeneity investigation.

When the material investigated by AFM indentation shows a certain degree of vertical heterogeneity (inclusions, layers, etc.), we consider the indented specimen as a composite structure, which contains deviations from homogeneous indentation behavior and standard contact mechanics models are failing to predict mechanical properties. To obtain information about the mechanical properties and position of the inclusion, we apply the contact mechanic model (for simplicity and calculation speed we considered the classical Hertz model in most of the cases) on several floating intervals along the indentation length. This procedure is acting similarly to a derivative of force vs. indentation, resulting in enhanced visualization of different mechanical behaviors, leading to a local apparent Young's modulus vs. indentation representation and demonstrating the heterogeneity and complexity of composite material. Analogous procedure was named stiffness tomography by Kasas and co-workers[47,48]. "AFMech Suite" software is equipped with the possibility to divide the indentation curves in intervals evaluating the apparent Young's modulus on a single point (spectroscopy), line (tomography), or entire map (hyperspectrum). Although this method is useful to qualitatively visualize mechanical properties of materials with different phases, the apparent Young's moduli can hardly represent the absolute moduli of single parts of the composite. Mechanical behavior of composites under indentation reflects the contribution of several factors such as geometries, mechanical convolution of single parts, and slippage at interface leading to highly complex mathematical problems and requiring a different approach for interpretation.

**FEM simulations**. A 2D axisymmetric numerical model was developed to study indentation on composite material system with a spherical indenter, mimicking AFM experiments as closely as possible. The total size of gel was 40 μm in lateral dimension (radius) and 43 μm in height, which is considered infinite if compared to indenting depth of several hundred nanometer to a few micrometer. The radius of the spherical indenter was set as in AFM experiments, $R = 2.5$ μm. During deformation, the left boundary, representing the symmetry axis, was allowed to move only vertically, while the bottom boundary was constrained to move horizontally. The right and top boundaries were not constrained, except when indenter is contacting the top objects (surface or inclusions): the contact area is restricted to follow the indenter contour ensuring hard contact between the indenter and the sample. Material properties for PS inclusions and indenter (silicon) are required as input parameters in the modeling and set as $E_{particle} = 3.3$ GPa, $\nu_{particle} = 0.34$, $E_{Si} = 160$ GPa, $\nu_{Si} = 0.22$. These hard materials have Young's moduli several orders of magnitude larger (GPa range) than hydrogels and bacteria (kPa range) in this study, showing negligible deformations. Young's moduli of hydrogel matrices were directly measured by AFM indentation, masking, and excluding inclusions surface area from analysis using "AFMech Suite", leading to $E_{hydrogel} = 30$ kPa, $\nu_{hydrogel} = 0.5$ for hard inclusions testing, and $E_{hard} = 27$ kPa and $E_{soft} = 6.5$ kPa for the supporting substrates of E. coli bacteria model. A methodological study, comparing AFM indentation and FEM model was recently published by our group[25].

Stress and deformation fields (example is reproduced in Supplementary Fig. 7) produced by indenter approaching vertically are directly converted in force (nN) vs. indentation (nm) to be compared with AFM approaching FCs. This comparison is performed mathematically on apparent Young's modulus vs. indentation, therefore dividing FC (both AFM and FEM) in intervals and apply the mechanical model on each of them. The discrepancy is calculated as $\Delta E$:

$$\Delta E = \frac{\sqrt{\sum_{i=1}^{N}\left(E_i^{FEM} - E_i^{AFM}\right)^2}}{N}, \quad (4)$$

where $E_i^{FEM}$, $E_i^{AFM}$, and $N$ represent apparent Young's modulus of $i^{th}$ interval from FEM, AFM, and the total number of intervals. We decided to use $\Delta E$ for comparison in order to equally evaluate differences near contact point and at high force with the same weight.

For unknown parameters such as bacteria modulus, position of inclusion, we run several simulations (7–10). The array of simulation is generally built varying a single unknown quantity around the most probable value guessed from AFM morphology and standard mechanical map. Loading the entire array of simulated FCs in "AFMech Suite", the FEM curve relative to the parameter that minimize $\Delta E$ is automatically selected and highlighted. An example of optimization procedure, varying modulus and minimizing $\Delta E$, is shown for bacteria on hydrogel system in Supplementary Fig. 8. Different mechanical responses due to interfacial bonding are playing an important role during determination of mechanical properties of single parts in composite. FEM model can be designed tuning the slip/friction at the interface between inclusion and matrix: two extreme conditions are represented by perfect bound (zero-slip) and perfect loose (free-slip). Although the deformation pattern around the inclusion at zero-slip and free-slip can be qualitatively similar (compression under particle and tension at edges), only quantitative analysis through AFM comparison can distinguish the proper behavior at the interface.

**Code availability**. For a complete and advanced data analysis of AFM FV data, we developed AFMech Suite, which is a software composed by five interacting interfaces in order to analyze complex systems. At the moment, import option requires Bruker or Asylum Research raw files or custom ASCII files, acquisition of and JPK and Keysight are under development. The software AFMech Suite v1.0 is freely available in GitHub open source repository (https://github.com/marsdeck/AFMech-Suite/archive/v1.0.zip) (version v1.0 available in Supplementary Software) along with a manual providing detailed explanations, procedural justifications, and practical examples. Future updated versions of AFMech Suite will be available on GitHub following the project at https://github.com/marsdeck/AFMech-Suite/.

Other free alternative applications are available in Java language, for example, WSXM[49] and Gwyddion[50] focused on imaging or OpenFovea[51] and AtomicJ[52] for FCs and mechanical analysis. The suite is written in Matlab language using event/object programming to provide an alternative tool for basic and advanced analysis. The analysis is in real time, allowing the user to be in control in each step of analysis and eventually interact (changing parameters, graphical representations) for user-friendly experience. After the analysis, the operator can export all graphical and numerical results using several output formats: image (.tiff), text (.txt), Matlab (.mat and .fig), Excel (.xls), and Origin (.opj). Moreover, the software can create custom metadata (.meta) to save time during investigation and quickly reload previous analysis. Interfaces of AFMech Suite are interacting in order to manipulate and visualize data where morphology, adhesion, and mechanical properties are often interconnected as in biological systems. The interfaces are described as:

1. Calibration: Visualize and perform operations about the calibration analysis of the cantilever, before or after mechanical tests. Analysis is focused in obtaining (1) elastic constant $K$ (N m$^{-1}$) from thermal noise measurements in air or liquid and (2) calculation of optical lever sensitivity Zsens (nm V$^{-1}$) from FC or FV on rigid, undeformable substrates (with negligible indentation). Interface allows the user to use advanced features such as: photodetector non-linearity correction and SNAP procedure[13], in order to produce high precision results.

2. Morphology: Visualize and perform operations on the raw height after acquisition. Raw morphology can be flattened removing sample tilt. A mask can be produced manually or with automatic height leveling to help in flattening or to exclude data from analysis in all interfaces. Real morphology (zero force topography) or indentation are available after mechanical analysis. Quantitative analysis is available using histograms and Gaussian fitting procedures.

3. Adhesion: FV analysis focused on retracting part of FCs. Adhesion analysis is important when the contact between probe and sample is adhesive and additional interaction forces play a role during indentation. Here, raw retracting FC can be aligned, pretreated in order to produce quantitative results. Adhesion analysis is especially useful when adhesive force is not negligible.

4. Mechanical: FV analysis focused on approaching part of FCs. Raw approaching FCs can be aligned and cleaned in order to find the contact point. Several contact mechanics models (depending on probe geometry and adhesion) can be used to produce quantitative Young's modulus analysis. Interacting mechanical analysis is especially useful when coupled with morphology for finite thickness correction or with adhesion for adhesive indentation problems.

5. Advanced: Advanced interface is a post-processing interface for the investigation of mechanical properties in a deeper way, for example, visualize profiles, Young's modulus tomography, perform a comparison with simulations or external files, and calculate the Young's modulus hyperspectrum.

Morphology, adhesion, and mechanical interfaces are conceptually divided in three parts: operation (on left), where raw AFM data are processed and corrected; quantification (right, top), where statistical analysis is performed through histograms and multi-Gaussian fitting; visualization (right, bottom), where final graphical results can be visualized and exported.

## Data availability

All data used in this manuscript are available from the authors on request.

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

## Acknowledgements

The authors would like to thank the National Science Foundation of China (21574086), Nanshan District Key Lab for Biopolymers and Safety Evaluation (No. KC2014ZDZJ0001A), Shenzhen Sci & Tech Research Grant (ZDSYS201507141105130), and Shenzhen City Science and Technology Plan Project (JCYJ20160520171103239) for financial support.

## Author contributions

M.G. and F.J.S. conceptualized this work; M.G. developed methodology; M.G. and Z.J. carried out AFM investigation; G.T. realized FEM model; M.G. wrote and developed analysis software; Z.J. tested and debugged software; C.S.B. synthesized nanocomposite

hydrogels and performed the rheological experiments; S.C. provided bacteria on
hydrogels specimens; M.G. wrote the original draft; all authors reviewed and edited the
manuscript; F.J.S. acquired funds and supervised this work.

## Additional information

**Competing interests:** The authors declare no competing interests.

