## [Peer Review File · Nature Communications]

Reviewers' comments:

Reviewer #1 (Remarks to the Author):

The manuscript concerns the development, calibration, and demonstration of the "AFMech Suite" program for characterizing mechanical properties of soft materials like gels. Whilst the authors have spent considerable efforts in constructing the software and discussing detailed data processing aspects, in my opinion the work is lacking the novelty and broad appeal to warrant publication in Nat. Comms.

This manuscript will be more suitable for consideration by a specialist journal in the field. Below are several suggestions/comments for consideration that may help to improve quality of their software:

The requirement to use the raw Bruker AFM file as input seems to be a major impediment in the current version of the software. It will be useful to be able to input load-displacement data as generic ASCII files.

Line 127: "The increase of stiffness resulted from the chain confinement of rubbers by filler aggregates"

Comment: Why filler aggregation was then neglected in the statistical distribution calculation?

Line 131: "shear modulus to E: $\nu=0.5$ "

Comment: Please clarify the reason to select $\nu=0.5$. Any evidence of incompressibility?

Line 151:

Comment: the idea of using the statistical distribution of randomly disperse spheres to evaluate the influence of the inclusions (i.e. the PS-sphere fillers) is interesting. However, since it is possible to locate the fillers laterally, what is the need for implementing the statistical method?

Line 178: "dividing the indentation curve in small intervals and fitting them using Hertz model"

Comment: It is not a good idea to use Hertz model since the sample is neither homogeneous nor without adhesive forces.

Line 256:

Comment: AFM nanoindentation on E. coli bacteria has already been well studied before. Meanwhile in line 274 "requiring FEM simulations for proper deconvolution for bacteria on soft substrate", I don't see the necessity for using a soft substrate that apparently affects the mechanical property measurement. Besides, the deconvolution of specimen on soft substrate has already been well-established by previous studies.

Reviewer #2 (Remarks to the Author):

This is a valuable contribution worthy of publication. In figures 1 and 3 the #2 is not visible enough and it should be pointed out more clearly that the AFM values not going up is a success, not a failure. I believe that using a standardized procedure such as the "standardized nanomechanical AFM procedure" (SNAP) is a good idea.

Reviewer #3 (Remarks to the Author):

The authors present a comprehensive tool for the mechanical characterization of biological species in the frame of hydrogel surfaces. The manuscript is well written and clearly explains the context

for which the tool is applied, with special emphasis on the theoretical models, the calibration procedure they have followed to test and validate the tool to obtain a generalizable protocol and the range of materials and composites for which it is applicable. Even the context for the application of the tool is somehow limited to certain physical characteristics of the samples under study, it is well defined and may have a wide range of applicability. Indeed, the authors are also offering a free software that includes all the tools used and presented in the manuscript. The software and manuscript are presented together with a comprehensive manual that covers the scope of application of the tool and the different practical aspects to use it.

Beyond the manuscript, the concept of open methodologies, specifically in the frame of determining mechanical properties at the nanoscale via atomic force microscopy is particularly appreciated by this referee: it is a common perception among the community that the availability of shared databases and generalized protocols on the determination of nanomechanical properties of biological materials should help to identify specific properties in biomolecules and cells opening the field to a faster and more reliable application of AFM as a nanomechanical tool in biological research areas.

In this sense, I consider the work well addressed and singular, pointing to the recognized needs of bionanomechanics community, so I believe it is worth to be considered for publication.

Atomic force microscopy and finite element simulations for heterogeneous soft-matter: methodology, software 'AFMech Suite' and biological applications

Response to Referees' Comments

Comments by Referee # 1

General Comment

The manuscript concerns the development, calibration, and demonstration of the "AFMech Suite" program for characterizing mechanical properties of soft materials like gels. Whilst the authors have spent considerable efforts in constructing the software and discussing detailed data processing aspects, in my opinion the work is lacking the novelty and broad appeal to warrant publication in Nat. Comms. This manuscript will be more suitable for consideration by a specialist journal in the field.

Response

We have chosen to submit our work to an open access journal targeted for a very broad audience, as we think that our work will benefit a whole host of research areas. Most obviously, soft matter specialists working on soft films and hydrogels will profit from it. Furthermore, biophysicists working on cells and/or bacteria will profit from accessing open access software and methodology with examples. Actually, we had exactly this discussion before submitting the work. I (Florian J. Stadler) always opted that we should go for a journal, which will be interesting for the general scientific public and not only for a small amount of specialized AFM scientists.

To us, this manuscript introduces a toolbox which a significant part of the scientific community will benefit from. While we present some data in the manuscript, we see these as verification for the software tools we introduce and thus as necessary

confirmation to prove and illustrate the capability of the AFMech Suite, which we developed for the benefit of the scientific community.

Specific Comment #1

The requirement to use the raw Bruker AFM file as input seems to be a major impediment in the current version of the software. It will be useful to be able to input load-displacement data as generic ASCII files.

Response

Yes, we are aware of the current limitation of AFMechSuite software with applicability restricted to raw files from Bruker software. At the moment, this limit is dictated by restricted access to raw data from other AFM companies, although we expect to broaden our software applicability after publication following the demands of AFM users in scientific community. We are currently working on this issue and planning a continuous and long-term support of software after initial release. So far, we prefer to work on importing raw data from the AFM company's software allowing the user to skip the intermediate ASCII step. In fact, force volume files must be converted following a proper format ordering to enable correspondence between morphology, mechanical properties and adhesion. In our opinion, it is better to have a dedicated import possibility for the most common raw data formats than to force users to convert their raw data to ASCII (of a certain file structure), which might have to be adjusted to allow for reimporting. This "modus operandi" increases the usability and reduces the sources of error, but at the same time we offer an ASCII-import option for all non-supported file types.

At this stage of revision, following the referee suggestion, we added the possibility of ASCII import plus we already add 'Asylum Research' import possibility. Based on AFM

user community we expect as future next steps to import 'JPK' and 'Keysight' raw data.

We uploaded a new version of software and a new documentation for software (Supporting Information 2_Software Guide, new section: Data Import) to guide user for 'Asylum Research' import data and creation of ordered ASCII files.

Specific Comment #2

Line 127: "The increase of stiffness resulted from the chain confinement of rubbers by filler aggregates"

Comment: Why filler aggregation was then neglected in the statistical distribution calculation?

The sentence "The increase of stiffness resulted from the chain confinement of rubbers by filler aggregates" was referred to the work of Alimardani et al.¹ from antecedent sentence. For the sake of clarity we modified the sentence and cited Alimardani et al. work again. In that work, volume filler concentration was $\phi_F = 24\%$ therefore confinement by filler aggregates was studied alongside surface interaction between polymeric chains and silica nanoparticle.

In our system aggregation of fillers can be neglected because concentration of filler is low ($\phi_F = 0.4$ wt.% for hydrogels in swollen state) and average particle inter-distance (see $\langle d \rangle$ Table 1) is almost 3 times the particle diameter, therefore showing a diluted composite system. Our goal, in fact, was to measure a single spherical filler without inter-particle interaction to highlight the capability of AFM coupled with FEM simulations.

Only exposed particles on surface were neglected from statistical analysis distribution because the modification of modulus sensed by AFM does depend on geometrical considerations as discussed in main text. In this way we use statistical analysis only to evaluate the average mechanical modulus from matrix phase and we show that only

for high indentation and small fillers AFM ‘starts’ to sense average mechanical properties like rheometer.

Specific Comment #3

Line 131: “shear modulus to E: $\nu=0.5$ ” Comment: Please clarify the reason to select $\nu=0.5$. Any evidence of incompressibility?

Response

Yes, PNIPAM based hydrogels behave as an almost ideal rubber and due to the incompressibility of both polymers and Newtonian liquids like water, it is safe to treat them as incompressible hyperelastic materials. Previous experimental works from our group comparing AFM indentation with rheology confirmed this Poisson’s ratio assumption,²⁻⁵ where Cox-Merz rule⁶ was used to convert shear into elongational modulus by $3|G^|=E$. Finally, previous direct rheological evidence demonstrate incompressible behavior.⁷*

As suggested, we added citations and specification in the main text.

Specific Comment #4

Line 151: Comment: the idea of using the statistical distribution of randomly disperse spheres to evaluate the influence of the inclusions (i.e. the PS-sphere fillers) is interesting. However, since it is possible to locate the fillers laterally, what is the need for implementing the statistical method?

Response

The theory about statistical distribution of randomly disperse spheres is applied to the bulk Young’s moduli values measured macroscopically by rheology. Therefore, statistical analysis is used when there is no access to local structures but only to averaged quantity, as is the standard for all mechanical experiments except AFM and nano-indentation. Statistical methods (histograms with Gaussian analysis) were used

in AFM analysis when the resolution, due to probe size and indentation level, is not enough to distinguish local features. In the paper, we show this effect of switching between local to averaged properties when probe size and indentation are bigger than local heterogeneity scale.

We point out that this question is particularly interesting in mechanobiology of cytoskeleton in living cells. Micrometric spherical probes are selected specifically to average local features on biomembranes such as glycocalyxes, brushes and membrane proteins, but maintaining enough resolution to focus on mechanical response of cytoskeletal layers (actin and microtubules). We added this discussion on main text.

Specific Comment #5

Line 178: “dividing the indentation curve in small intervals and fitting them using Hertz model”

Comment: It is not a good idea to use Hertz model since the sample is neither homogeneous nor without adhesive forces.

Response

Yes, Hertz model is the most simple approximation of contact mechanics when using spherical indenter. In Supporting Information we showed the analysis of adhesive interaction probe sample as well a discussion regarding the contact mechanics model used in this study.

Although adhesion is small and can be neglected when fitting the whole indentation range, adhesion can influence intervals especially near the surface (shallow indentation). However, as our measurements were performed under water, the influence of adhesion is minute, as was shown in Supporting Information (SI.5 Adhesion analysis) and in a recent paper from our group,⁵ on the basis of a comparison between approaching and retraction curves.

In order to clarify this, we point out in the text that the procedure of ‘dividing in small intervals’, similarly to ‘stiffness tomography’, is not mathematically correct in principle and no absolute value of Young’s modulus can be quantified (we always refer to Apparent Young’s Modulus throughout the text). Young’s modulus vs. Indentation is only qualitative and acting as a derivative of force with indentation. It was used specifically to compare single AFM experimental curves with an FEM array of simulations in order to produce quantitative results weighting all indentation intervals in the same way (comparison using Force vs. Indentation curves is more sensitive to data at high force/indentation). In our opinion the most important point is the statement that when comparing of AFM and FEM in the same way, a minimization algorithm will lead to the simulation nearest with experimental data.

As a demonstration of ‘daily operation’ basis we preferred Hertz model because can be expressed in mathematical closed form leading to 50 times faster performance than more precise models without closed form (for example Sneddon spherical or JKR models). With AFMech Suite, users can select a different models for comparison if they require (AFMech Suite automatically highlights if long time is expected for operation). Furthermore, due to the way how the models are implemented into the program it is relatively easy to add additional models as needed.

Specific Comment #6

Line 256: Comment: AFM nanoindentation on E. coli bacteria has already been well studied before. Meanwhile in line 274 “requiring FEM simulations for proper deconvolution for bacteria on soft substrate”, I don’t see the necessity for using a soft substrate that apparently affects the mechanical property measurement. Besides, the deconvolution of specimen on soft substrate has already been well-established by previous studies.

Response

The whole point of this part in the paper is not the measurement of the modulus of bacteria. It is rather to prove that it is possible to deconvolute the properties of the substrate and the sample, which for biological applications will play a larger role.

While there are several reports of AFM mechanics of cancer and healthy cells residing on hydrogels surfaces, only few studies were reported about AFM on bacteria attached to hydrogels. Moreover, in those studies, AFM is used in retraction mode focusing on adhesion force between gel surface and bacterial membrane. In our study, we deconvolute the moduli of the two phases (hydrogel + bacterium) with the help of FEM simulations investigating the influence of substrate stiffness on bacteria and the strength of bound at the interface. The substrate is only influencing the apparent modulus of bacteria, while there is no measureable influence on bacterial structural reorganization due to mechanical response. This is certainly not true for living cells, reorganizing the cytoskeletal structure to adapt to extra cellular matrix and, furthermore, in some cases it is known that the modulus of the surroundings influence the specialization a stem cell develops.

Therefore, we propose an instrument, such as the proposed methodology and AFMech Suite can be applied to retrieve real modulus of living cells without influence of substrate.

Using complex multi-layer models or finite element simulations for deconvolution are strategies already available in literature. The novelty of our work resides in the guidance of FEM simulations through well-calibrated (SNAP method) AFM measurements (morphology or moduli of parts unaffected by convolution), a generation of arrays of simulations in order to optimize unknown parameters directly comparing 'Apparent Young's moduli' experimental data.

Comments by Referee # 2

General Comment

This is a valuable contribution worthy of publication. In figures 1 and 3 the #2 is not visible enough and it should be pointed out more clearly that the AFM values not going up is a success, not a failure. I believe that using a standardized procedure such as the "standardized nanomechanical AFM procedure" (SNAP) is a good idea.

Response

Following the reviewer suggestions, we increased visibility of #2 in all figures and we discussed more clearly about behavior of Young's modulus.

Comments by Referee # 3

General Comment

The authors present a comprehensive tool for the mechanical characterization of biological species in the frame of hydrogel surfaces. The manuscript is well written and clearly explains the context for which the tool is applied, with special emphasis on the theoretical models, the calibration procedure they have followed to test and validate the tool to obtain a generalizable protocol and the range of materials and composites for which it is applicable. Even the context for the application of the tool is somehow limited to certain physical characteristics of the samples under study, it is well defined and may have a wide range of applicability. Indeed, the authors are also offering a free software that includes all the tools used and presented in the manuscript. The software and manuscript are presented together with a comprehensive manual that covers the scope of application of the tool and the different practical aspects to use it.

Beyond the manuscript, the concept of open methodologies, specifically in the frame of determining mechanical properties at the nanoscale via atomic force microscopy is particularly appreciated by this referee: it is a common perception among the community that the availability of shared databases and generalized protocols on the determination of nanomechanical properties of biological materials should help to identify specific properties in biomolecules and cells opening the field to a faster and more reliable application of AFM as a nanomechanical tool in biological research areas.

In this sense, I consider the work well addressed and singular, pointing to the recognized needs of bionanomechanics community, so I believe it is worth to be considered for publication.

Response

Thank you for the appreciation of our work.

References:

- 1 Alimardani, M., Razzaghi-Kashani, M. & Ghoreishy, M. H. R. Prediction of mechanical and fracture properties of rubber composites by microstructural modeling of polymer-filler interfacial effects. *Mater Design* **115**, 348-354, doi:10.1016/j.matdes.2016.11.061 (2017).
- 2 Tang, G., Galluzzi, M., Biswas, C. S. & Stadler, F. J. Investigation of micromechanical properties of hard sphere filled composite hydrogels by atomic force microscopy and finite element simulations. *J Mech Behav Biomed Mater* **78**, 496-504, doi:10.1016/j.jmbbm.2017.10.035 (2018).
- 3 Wang, Q. *et al.* Random copolymer gels of N-isopropylacrylamide and N-ethylacrylamide: effect of synthesis solvent compositions on their properties. *Rsc Adv* **7**, 9381-9392, doi:10.1039/C6RA27348C (2017).
- 4 Biswas, C. S. *et al.* Versatile Mechanical and Thermo-responsive Properties of Macroporous Copolymer Gels. *Macromol Chem Phys*, 1600554-n/a, doi:10.1002/macp.201600554 (2017).
- 5 Galluzzi, M. *et al.* Space-resolved quantitative mechanical measurements of soft and supersoft materials by atomic force microscopy. *NPG Asia Materials* **8**, e327, doi:10.1038/am.2016.170 (2016).
- 6 Cox, W. P. & Merz, E. H. Correlation of Dynamic and Steady Flow Viscosities. *Journal of Polymer Science* **28**, 619-622 (1958).
- 7 Stadler, F. J., Friedrich, T., Kraus, K., Tieke, B. & Bailly, C. Elongational rheology of NIPAM-based hydrogels. *Rheologica Acta* **52**, 413-423, doi:10.1007/s00397-013-0690-x (2013).

REVIEWERS' COMMENTS:

Reviewer #2 (Remarks to the Author):

I believe that the authors adequately addressed the questions and comments of reviewer #1. I believe the manuscript is now suitable for publication.

Reviewer #3 (Remarks to the Author):

The comments to the referees were properly addressed. I recommend to consider it for publication